# Effect of Polyethylene Glycol-Simulated Drought Stress on Stomatal Opening in “Modern” and “Ancient” Wheat Varieties

**DOI:** 10.3390/plants13111575

**Published:** 2024-06-06

**Authors:** Ilva Licaj, Anna Fiorillo, Maria Chiara Di Meo, Ettore Varricchio, Mariapina Rocco

**Affiliations:** 1Department of Science and Technology, University of Sannio, 82100 Benevento, Italy; ilva.licaj@unisannio.it (I.L.); mardimeo@unisannio.it (M.C.D.M.); etvarric@unisannio.it (E.V.); 2Department of Biology, University of Rome Tor Vergata, 00133 Rome, Italy; anna.fiorillo@uniroma2.it

**Keywords:** drought stress, leaf, Saragolla cultivar, Svevo cultivar, wheat

## Abstract

Climate change is leading to an increase in the intensity, duration, and frequency of severe droughts, especially in southern and southeastern Europe, thus aggravating water scarcity problems. Water deficit stress harms the growth, physiology, and yield of crops like durum wheat. Hence, studying ancient wheat varieties’ stress responses could help identify genetic traits to enhance crop tolerance to environmental stresses. In this background, this study aimed to investigate the effects of PEG 6000-stimulated drought stress in the ancient wheat variety Saragolla and the modern one Svevo by analyzing various biochemical and molecular parameters that can especially condition the stomatal movement. Our data revealed that drought stress caused a significant increase in the levels of total soluble sugars, ABA, and IAA in both selected cultivars to a greater extent in the Saragolla than in the Svevo. We demonstrated that, under water deficit stress, calcium dynamics as well as the expression of *ERF109*, *MAPK3/6*, *MYB60*, and *TaTPC1*, involved in the activation of drought-related calcium-sensitive pathways, display significant differences between the two varieties. Therefore, our study provided further evidence regarding the ability of the ancient wheat variety Saragolla to better cope with drought stress compared to the modern variety Svevo.

## 1. Introduction

Climate change is gradually leading to reduced precipitation and higher global average temperatures, enhancing the severity of droughts and threatening global food security [1,2,3,4]. Increased drought has a significant impact on plant growth and development, resulting in a notable decline in cereal yields, including durum wheat (*Triticum turgidum* ssp. *durum*), which is one of the most important cereal crops grown worldwide [5]. In this framework, the rediscovery of ancient varieties with their wide genetic diversity is a promising strategy to improve wheat fitness in light of climate change. Indeed, despite their low yields, ancient wheat varieties are suitable for extensive farming systems under stress conditions where conventional wheat cultivars fail to thrive [6]. Among these ancient varieties, Saragolla, a cultivar grown in the southern region of Italy, represents a good candidate for identifying genetic traits that could optimize plant productivity under drought stress conditions.

In response to drought stress, plants undergo a series of morphological, physiological, and biochemical changes, including improvements in root and leaf architecture, stomatal closure, and the regulation of water content and osmotic balance [7]. Osmotic adjustment primarily occurs through the accumulation of low-molecular-weight compounds such as heat shock and LEA proteins, amino acids, and sugars, which play crucial roles in desiccation tolerance. Soluble sugars, particularly sucrose, help maintain cellular turgor, scavenge reactive oxygen species (ROS), and act as signaling molecules for various tolerance-related mechanisms, including abscisic acid (ABA)-induced stomatal closure [8,9,10,11]. Stomatal closure represents one of the earliest responses of plants to drought stress [12]. This process is regulated by an intricate network of signaling pathways, with ABA, ROS, and Ca^2+^ serving as key regulators. In particular, ABA accumulation in guard cells activates many downstream Ca^2+^-mediated pathways, involving calcium-dependent protein kinases (CDPKs) and calcium-binding proteins [13,14]. These pathways, along with the gene expression reprogramming, are critical for activating ion channels, such as S-type anion channels and inward-rectifying potassium channels, leading to ion and water efflux from guard cells, inducing stomatal closure and reducing transpirational water loss [15]. Thereby, in this study, we shifted our attention to the association between guard cells’ closure and intracellular Ca^2+^ concentration, considering the expression of several protein kinases and phosphatases as signal transducers [16,17,18,19,20]. Based on these, we investigated *ERF109*, *MAPK3/6*, *MYB60*, and *TaTPC1* gene expression since they are involved in the activation of calcium-sensitive response pathways under drought stress and indirectly in stomatal movement.

Although ABA is known to be the main and most characterized hormone involved in stomatal closure, it is equally well known that the regulation of stomatal movements results from fine crosstalk between phytohormones, including jasmonates (JAs), ethylene, auxins, and cytokinins [21,22,23]. In general, ABA and JA are considered positive regulators of stomatal closure, while auxin and cytokinins regulate stomatal opening [12]. Indole-3-acetic acid (IAA), in particular, typically associated with plant growth and development, has also been proposed as a crucial regulator of drought stress-signaling pathways [24]. Despite this, the precise mechanisms underlying the involvement of IAA in drought stress responses and its interconnections with other hormones remain to be fully elucidated [24]. It has been demonstrated that under drought stress conditions, a rearrangement of auxin levels occurs, with an increase in roots and a decrease in leaves [25,26]. This IAA redistribution can help plants perceive changes in water availability in the soil [27] and modulate ABA signaling pathways, thereby influencing stomatal behavior. In *Arabidopsis thaliana*, IAA antagonizes ABA-induced stomatal closure through the downregulation of ABA-responsive genes [28]. Therefore, a full understanding of these molecular mechanisms and the identification of genetic markers typical of stress-tolerant varieties are imperative for improving crop characteristics, including tolerance to drought stress.

Previous studies revealed that the ancient wheat genotype Saragolla copes better with drought stress compared to the modern one Svevo, displaying an improved water use efficiency, reduced oxidative damages, and higher antioxidant enzyme activities, as well as in the regulation of stomatal movement [29,30,31]. In this study, we further investigated the molecular mechanisms leading to the increased drought tolerance of the Saragolla variety. The responses of Saragolla seedlings to a polyethylene glycol (PEG-6000)-simulated drought stress were compared to the traditional variety Svevo. First, we analyzed the accumulation of soluble sugars in response to PEG-6000-induced drought stress, as one of the main mechanisms involved in tolerance acquisition. Then, we examined the molecular mechanisms involved in the regulation of stomatal movement in the two cultivars by determining ABA and IAA levels, intracellular Ca^2+^ dynamics, and the expression of *ERF109*, *MAPK3/6*, *MYB60*, and *TaTPC1*, which are signal transducers of the calcium-sensitive osmotic stress response and, indirectly, of stomatal movement.

## 2. Results

### 2.1. Changes in Concentrations of Solutes under Drought Stress

The accumulation of compatible solutes represents an essential strategy employed by plants to protect themselves and survive under drought-stress conditions [32]. Since drought-tolerant cultivars have higher soluble sugar content compared to non-tolerant ones, and a direct correlation between sugar accumulation and tolerance to abiotic stresses has been observed [33], we compared total soluble sugars and sucrose accumulation in Saragolla and Svevo varieties in response to PEG-6000-induced drought stress. Figure 1 shows that, under physiological conditions, the concentrations of both parameters were comparable in the two cultivars. Accordingly, under drought stress conditions, a significant increase in the levels of total soluble sugars and sucrose content was recorded in both genotypes. Notably, the increase was more pronounced in the ancient variety Saragolla, which exhibited higher accumulations of soluble sugars by approximately 28.2% and sucrose by approximately 34.9% relative to the modern wheat cultivar.

### 2.2. Analysis of the Contents of Endogenous Hormones under Water Deficit Treatments

Phytohormones are essential regulators of numerous plant growth and development processes, exerting their fundamental action under both physiological and stress conditions [21,23]. Hence, the modulation of phytohormone levels and their crosstalk are among the most important processes for acquiring tolerance to environmental stresses. For instance, fine-tuning between ABA and auxin levels plays a key role in regulating stomatal movement in response to water stress [24]. ABA and IAA levels were determined under control and drought stress in Saragolla and Svevo seedlings (Figure 2). The results demonstrated that, under well-watering conditions, no difference in the amounts of ABA and IAA was observed. In contrast, PEG-6000 treatment induced a significant increase in ABA and IAA content in both wheat cultivars, although to a slightly greater extent in the Saragolla variety. Indeed, Figure 2A shows that under drought stress, ABA increased by about 56.9% in Saragolla and 49.2% in Svevo, while IAA, as shown in Figure 2B, increased by approximately 37.8% in Saragolla and 24.4% in Svevo compared to control conditions.

### 2.3. Confocal Interpretation and Signal Transduction

In response to drought stress, the activation of calcium signaling is pivotal in enhancing plant tolerance [34]. As a crucial second messenger, Ca^2+^ regulates numerous physiological and biochemical processes in response to stresses, including the protection of cell membranes, the scavenging of reactive oxygen species (ROS), the biosynthesis of defense hormones, and the regulation of stomatal movements [34].

The Ca^2+^ spatial dynamics were analyzed in the leaf mesophyll after 5 days of PEG-6000 treatment using the fluorescence of the Ca^2+^ probe Fluo-4. As shown in Figure 3, under normal watering conditions, the fluorescence signal in the cell wall and cytoplasm was lower than under stress treatment, and almost no fluorescence was observed in the cytoplasm, especially in the Saragolla control. In contrast, for both cultivars subjected to osmotic stress, the stress-induced Ca^2+^ oscillations resulted in a significant increase in the fluorescence signal (Figure 3B,D,F,H). This event is illustrated in the graphs of Figure 3B,D,F,H where the calcium signals under osmotic stress were higher at some pixels compared to those without stress. The profiles of the yellow lines drawn in Figure 3a–d represent the values of fluorescence intensity (green line) at each pixel point with or without PEG-6000 treatment (Figure 3A,C,E,G). The effect of PEG-6000 on Ca^2+^ dynamics was more pronounced in the cultivar Svevo than in Saragolla. In fact, in Svevo, a greater change in Ca^2+^ distribution and a higher increase in fluorescence signal intensity were observed in all cell types, including guard cells.

To understand whether the alteration of Ca^2+^ dynamics also affected the downstream signals, the expression of *ERF109*, *MAPK3/6*, *MYB60*, and *TaTPC1*, as transducers of the calcium-mediated osmotic stress response, was analyzed. Figure 4 demonstrates that, despite *MAPK3/6* levels being comparable between the two cultivars under control conditions, PEG-6000 treatment induced a significant activation of both *MAPK3* and *MAPK6* expression in Saragolla, while no substantial difference was observed in Svevo. Regarding *ERF109* and *TaTPC1* (Figure 4), treatment with PEG-6000 did not affect their expression in either cultivar, which remained almost the same as control conditions, suggesting the non-responsiveness of these genes to osmotic stress. However, it is interesting to note that the expression of these genes is very different between the two cultivars under control conditions, with significantly higher expression levels in Svevo than in the Saragolla cultivar. Furthermore, the expression of the *Triticum* transcription factor *MYB60* was analyzed. In physiological conditions, *MYB60* expression was higher in the Svevo variety than in Saragolla. Under water stress, no alteration of *MYB60* transcription was observed in Saragolla while a significant reduction occurred in Svevo, resulting in *MYB60* expression levels even lower than in the other variety.

## 3. Discussion

Understanding the biochemical, physiological, and molecular mechanisms activated under limited water availability is of great importance for developing high-yield, drought-tolerant wheat varieties and maintaining agriculture productivity. On this basis, in this work, we continued to delineate the differences in drought stress responses of two different wheat genotypes: Saragolla, an ancient cultivar, and Svevo, an elite variety widely used in intensive agriculture. These two cultivars are already known to differ in their growth and drought stress response, with the ancient cultivar Saragolla previously reported to be more drought-resistant [29,30,31]. This makes Saragolla a potential reservoir of genetic diversity that can be exploited to select tolerance traits.

In our previous research [29,30,31], the comparison of morpho-physiological traits and biochemical responses of Saragolla and Svevo varieties to PEG-6000-induced drought stress demonstrated that Saragolla accumulated less hydrogen peroxide (H_2_O_2_). Consequently, the oxidative damage, determined by quantifying Malondialdehyde (MDA) production and ion leakage, was less severe in this cultivar compared to Svevo. Moreover, Saragolla accumulated more proline in the roots and maintained a higher photosynthetic and transpiration rate than Svevo, after 5 days of the osmotic stress imposition, thus contributing to the maintenance of the plant’s growth and development rate. In addition, we extensively studied the stomatal characteristics, revealing that drought stress also affected the morphology and degree of stomatal opening differently in the two varieties. Overall, this study provided the first evidence of Saragolla’s superior adaptation to drought stress compared to the Svevo genotype. Hence, in this work, to gain deeper insight into the effects of genotypic variations on drought tolerance, we expanded our studies by examining other parameters in wheat leaves under the PEG-6000 treatment.

An elevated accumulation of reduced forms of water-soluble sugars or sugar alcohols is essential for cellular protection against dehydration [35], ensuring structural stability, osmotic balance, ROS scavenging, metabolic maintenance, and water retention. It is well established that under water stress conditions, the accumulation of compatible solubles in leaves and roots [36,37] is crucial for osmotic adjustments [38] and maintaining cellular turgor pressure, thereby reducing water loss. On this basis, one of the objectives of our work was to understand whether the different responses to water stress previously observed [29,30,31], such as reduced levels of ROS and oxidative stress, higher energy production, and the ability to preserve available water resources, could be ascribable to different accumulations of soluble sugars between the two varieties. Our data revealed a significant increase in soluble sugars and sucrose content under PEG-6000 treatment in both wheat genotypes, with a more pronounced effect in the ancient variety Saragolla, indicating that sugars contribute to the improved drought stress tolerance of this variety [38,39]. Besides their function in preventing cellular dehydration, soluble sugars also act as stabilizers of membranes and cellular enzymes, and as supporters of growth, photosynthesis, and reproduction [40]. Hence, it is conceivable that there is a direct link between the increase in soluble sugars and the higher photosynthetic rate, root/shoot ratio, and leaf area [29,30] observed in Saragolla under drought stress conditions.

Other important players in abiotic stress tolerance are phytohormones [41]. Among the drought-adaptation strategies, a prominent role is played by the phytohormone-mediated regulation of stomatal movement [42], which prevents water loss and plant dehydration [43]. Stomatal closure is positively regulated by ABA and JA, and negatively regulated by IAA and cytokinins [12]. ABA, in particular, is generally recognized as the most important player in drought stress resistance [44,45], and under stress conditions, its concentration rapidly increases [46]. Our data demonstrated that under drought stress, ABA content significantly increased in both wheat genotypes, suggesting that the drought-sensitive phenotype observed in the Svevo cultivar cannot be ascribable to an impaired ability of this variety to synthesize and accumulate ABA. Besides ABA, it has been recently demonstrated that the regulation of auxin homeostasis is essential in drought stress responses [47]. Auxins are generally considered to be negative regulators of drought tolerance, and in wheat leaves and rice, reduction in IAA levels promotes the activation of several tolerance-related processes, such as the transcriptional activation of genes coding for Late Embryogenesis Abundant (LEA) proteins [48]. Despite this, in some plant species, a transient increase in IAA levels has been observed during the early stages of stress, which then undergoes an abrupt decrease during the acclimation phase [48]. Increasing the concentration of IAA induces water uptake in protoplasts [49], a process generally associated with the opening of stomata; for this reason, IAA is considered an antagonist of ABA [50,51]. Our results demonstrated an increase in IAA concentration for both Saragolla and Svevo cultivars; however, a higher percentage increase compared to the control is observed in Saragolla compared to Svevo. Given that ABA content increased to similar levels in both treated cultivars, the substantial rise in IAA concentration in Saragolla is likely a contributing factor to the partial opening of its stomata. These data are in agreement with our previous results showing that under drought stress, stomatal closure occurred in Svevo, while in Saragolla, the stomata remained partially open [31]. The evidence that water stress is unable to induce the complete stomatal closure in Saragolla prompts us to hypothesize that other mechanisms, such as the greater accumulation of sugars and proline and reduced membrane damage [29], contribute to the maintenance of optimal ψ and hence to the greater tolerance of the Saragolla cultivar compared to the Svevo.

Stomatal closure is triggered by various second messengers, among which calcium (Ca^2+^) plays a pivotal role [20,52]. In response to osmotic stress conditions, Ca^2+^ channels open, releasing Ca^2+^ from intracellular stores and, as a result, leading to its accumulation both in the cytosol and in the chloroplasts. The transient increase in intracellular Ca^2+^ concentration and the subsequent activation of downstream signaling cascades are essential for the induction of stomatal closure [14]. To understand whether the difference in stomatal opening between Saragolla and Svevo under drought stress could be ascribable to different concentrations and the spatial distribution of Ca^2+^ ions, Ca^2+^ dynamics in response to drought stress were analyzed using the Fluo-4 fluorescent probe. Our results demonstrated that in response to PEG-6000 treatment, increased Ca^2+^ levels were observed in both cultivars. Despite this, Saragolla showed a faint fluorescence compared to Svevo, meaning a lower calcium concentration in response to stress in this variety. Therefore, it is possible to speculate that the inability of Saragolla stomata to be in a fully closed state may be a consequence of altered Ca^2+^ signaling.

The increase in cytoplasmic Ca^2+^ governs the phosphorylation of different *MAPKs* that, on their side, can regulate the expression of transcription factors (TFs) involved in responses to osmotic stress [53]. Despite the strong relationship between the activation of *MAPK* and Ca^2+^ signaling, it is also known that excessively high stromal Ca^2+^ concentration can inhibit this process [53]. This phenomenon may justify the increased expression of *MAPK3/6* observed in Saragolla compared to Svevo, where Ca^2+^ levels remain lower than in the other variety. Downstream of *MAPK* activation, many TFs are involved in drought stress responses. In particular, since the release of the *MYB* and *ERF* TFs triggers chloroplast-to-nucleus signaling required for the regulation of stomatal movements [54], the expression of two representative TFs, *ERF109* and *MYB60*, was investigated. Our data showed that although water stress does not result in the activation of *ERF109* in both varieties, the expression of this gene was higher in Svevo than in Saragolla under both control and stress conditions. As far as *MYB60*, it is an essential transcriptional regulator involved in monitoring water availability and fine-tuning plant responses to drought stress, including stomatal movements [55,56]. Under severe drought stress conditions, indeed, the downregulation of *MYB60* induces stomatal closure [56]. Our data reveal that drought stress significantly inhibited *MYB60* expression in Svevo, while no significant difference was observed in Saragolla. The inhibition of *MYB60* expression is in agreement with the reduction in stomatal pores previously observed in Svevo by Licaj et al. [31], confirming that in response to water deprivation, Svevo is more prone to closing stomata compared to Saragolla, where the expression of *MYB60* was not affected.

Ultimately, we analyzed the expression of *TaTPC1*, which encodes a Ca^2+^-permeable channel required for facilitating Ca^2+^ entry into plant cells and, consequently, inducing stomatal closure [57,58,59,60]. Our results showed that *TaTPC1* expression was not affected by drought stress in either Saragolla or Svevo, suggesting that this gene is not responsive to drought stress. Nevertheless, the expression of *TaTPC1* was significantly higher in Svevo, both under control and stress conditions, than in Saragolla, supporting the hypothesis that the increased transcription of *TaTPC1* may likely accelerate stomatal closure in the presence of Ca^2+.^

Based on the obtained results, we developed a schematic illustration of the molecular mechanisms activated by the two wheat varieties in response to drought stress (Figure 5).

## 4. Materials and Methods

### 4.1. Plant Materials and Treatments

The experimental design and hydroponic growth conditions were set according to Licaj [29,30,31]. Seeds of the *Triticum turgidum* ssp. *durum* cultivars Svevo (Agrisemi Minicozzi, Benevento, Italy) (“modern” wheat) and Saragolla (provided by a farm located in South Italy) (“ancient” wheat) were surface-sterilized in 20% sodium hypochlorite for 20 minutes. The germinated seeds were selected to grow in a hydroponic tank with a 1/2 Hoagland solution in a climate room, 50% humidity, with a photoperiod of 16 h of light at 24 °C and 8 h of darkness at 18 °C. One week after the germination, the seedlings were randomly divided into two groups. For osmotic stress (treatment group), seedlings were treated for five days by adding the solution polyethylene glycol (PEG-6000) to a final concentration of 18% (*w*/*v*), while the untreated group (control) was grown for the same period without PEG-6000 treatment. Leaves from both control and water-deprived plants were collected at the end of the PEG-6000 treatment.

### 4.2. Measurement of Soluble Sugar and Sucrose Content

Total soluble sugar content was measured following the method described by Irigoyen et al. [61]. One hundred milligrams of leaves were homogenized in liquid nitrogen and interfering pigments were extracted with 100% acetone. Samples were then incubated with 5 mL of 80% ethanol and centrifuged at 5000× *g* for 10 min. This extraction procedure was repeated three times. In total, 1 ml of the resulting extracts was added to 5 mL of a 0.2% anthrone solution. The mixture was heated at 100 °C for 10 min and then cooled on ice for 5 min. Total soluble sugars were determined using a spectrophotometer at 630 nm. A standard curve was constructed using glucose. Sucrose content was measured by using the resorcinol method and estimated based on the absorbance at a wavelength of 480 nm and a standard curve [62].

### 4.3. Determination of ABA and IAA Contents

The extraction of ABA and IAA and subsequent HPLC analyses were conducted following the method outlined by Manzi et al. [63], with minor adjustments. Briefly, 1.0 g of leaves was frozen in liquid nitrogen and then homogenized using a mortar and pestle. Next, 2.5 mL of methanol as the extraction solvent was added. The mixture was then vortexed vigorously to ensure thorough mixing and centrifuged at 16,000× *g* for 10–15 min at 4 °C. The supernatant was concentrated under vacuum to one-tenth of its original volume. Pure water (pH 9) was added, and the sample was extracted with ethyl acetate. After centrifugation at 16,000× *g* for 2 min, the aqueous phase (adjusted to pH 3) was re-extracted with ethyl acetate. The organic phase was dried under vacuum and dissolved in 30 μL of methanol for the HPLC analysis.

The HPLC analysis was conducted using an LC-20 Prominence HPLC system (Shimadzu, Kyoto, Japan) featuring an LC-20AT quaternary gradient pump, SPD-M20A photodiode array detector (PDAD), and SIL-20 AH autosampler with a 20 μL injection volume. Plant hormones were separated on a Gemini-NX C18 column (250 × 4.5 mm, 5 μm particle size) (Phenomenex, Torrance, CA, USA), equipped with a Security Guard^®^ pre-column (Phenomenex). The separation employed a gradient of acetonitrile with 0.1% (*v*/*v*) trifluoroacetic acid in aqueous 0.1% (*v*/*v*) trifluoroacetic acid at 45 °C. The acetonitrile concentration was ramped from 15 to 30% over 5 min, from 30% to 50% over 5 min, and from 50% to 80% over 2 min, and then the starting elution conditions were restored, at a flow rate of 1.5 mL/min. Compounds were identified by comparing their retention times and UV spectra to IAA (12886, Sigma, St Louis, MO, USA) and ABA (A1049, Sigma) standards, which were also used to create calibration curves (1–100 μg/mL) at specific wavelengths (λIAA = 254 nm; λABA = 254 nm). Results are expressed as μg of hormone per gram of fresh tissue. Data from independent assays were statistically analyzed with mean values ± SD of three independent extractions reported [64].

### 4.4. Confocal Settings and Image Processing

The fluorescent calcium ion probe Fluo-4 (F10489, Thermofisher Scientific, Waltham, MA, USA) was used to determine the Ca^2+^ level in wheat leaves under control and osmotic stress conditions, as previously reported by Jing et al. [14]. The abaxial surface of Saragolla and Svevo leaves was placed on transparent tape, and the upper epidermis was removed using a surgical blade. The samples were then incubated in Hank’s balanced salt solution (without calcium ions) containing 20 μmol/L Fluo-4 for 40 min at 25 °C in the dark. Subsequently, the leaves were washed 3–5 times with Hank’s buffer to remove the excess Fluo-4 and then incubated again at room temperature for 20 min in the same solution to ensure the complete dissolution of the esterification probe. Finally, the samples were placed on a glass slide with 0.5 mL of Hank’s buffer to complete the slice preparation. The Ca^2+^ levels were examined using a confocal laser scanning microscope (AXR Nikon Confocal Microscope, Nikon, Japan). Fluorescence intensity was detected by setting the instrument as follows: excitation—488 nm, and the emission in the range of 505–544 nm; Pinhole—34.2 μm; DG—581; AO—0.1; and AG—1.34. Contrast and brightness (laser detection intensity), laser scanning area, and position in the leaves were adjusted in the same manner for all images. The fluorescence intensity was analyzed using NIS-Elements AR v.5.20.00 Software.

### 4.5. PCR (qRT-PCR) Analysis

To analyze the expressions of *TaTPC1*, *ERF109*, *MYB60*, and *MAPK3/6* under PEG-6000 treatment and normal water conditions, the total RNA was extracted from 12-day-old Saragolla and Svevo leaves using “Spectrum Plant Total RNA Kit” (Sigma-Aldrich, Milan, Italy), according to the manufacturer’s instructions. RNeasy/QIamp columns and RNase-Free DNase set (Qiagen, Milan, Italy) were used to degrade genomic DNA and obtain an eluate of pure RNA. The RNA was then reverse-transcribed to cDNA using “ImProm-II Reverse Transcription System Kit” (Promega, Milan, Italy). Gene-specific primers regarding *ERF109*, *MAPK3/6*, and *MYB60* (as shown in Table 1) were designed using the NCBI Primer Blast tool, whereas the primers for *TaTCP1* were previously designed and tested by Wang et al. [57]. An “EvaGreen 2X qPCR MasterMix-R” kit (Applied Biological Materials, Vancouver, BC, Canada) was used for qRT-PCR. The thermal cycler using the “7300 Real-Time PCR System” was set to perform an initial denaturation at 95 °C for 1 min, an annealing phase of 5 min at 95 °C, and 40 successive cycles of denaturation (95 °C for 30 s), annealing (60 °C for 30 s), and extension (72 °C for 30 s). mRNA levels were normalized to actin (*β-Actin*). Experiments were carried out in triplicate and the relative quantification in gene expression was determined using the 2^−∆∆Ct^ method [65].

### 4.6. Statistical Methods

For comparisons between the two groups, statistical significance was calculated using Student’s *t*-test (Graph Pad Prism 5 software). Bars are represented as the standard deviation of the mean (SD). * *p* < 0.05 was considered statistically significant.

## 5. Conclusions

Stomatal behavior in plants, including wheat cultivars, is regulated by a complex network of signals that interact to manage water loss and gas exchange, particularly under drought stress, such as hormonal cues, Ca^2+^, reactive oxygen species (ROS), and other secondary messengers. In tolerant wheat cultivars, these signals integrate to modulate stomatal aperture, ensuring water efficiency while minimizing the impact on photosynthesis. Based on that, in summary, our results provide further evidence of Saragolla’s superior adaptation to drought stress, offering insights into the intricate mechanisms that can inform the development of improved drought-tolerant wheat varieties. This is crucial for ensuring food security in the face of climate change-induced water scarcity.

Our investigation into biochemical parameters revealed a significant increase in soluble sugars and sucrose content under drought stress, with Saragolla showing a more pronounced effect. This indicates that sugars contribute to the improved drought stress tolerance of Saragolla, acting as osmoprotectants and supporting essential cellular functions. Additionally, our data showed a substantial increase in ABA and IAA levels in both cultivars under drought stress, with Saragolla exhibiting a higher percentage increase in IAA concentration compared to Svevo. In the majority of studies, this is associated with stimulated stomata opening and is considered an ABA antagonist, likely contributing to the partial stomata opening observed in Saragolla under drought stress. Furthermore, our study highlights the complex interplay between calcium concentration, *MAPK* phosphorylation, and transcription factor regulation in stomatal movement under osmotic stress. Differences in calcium concentration between Saragolla and Svevo suggest altered Ca^2+^ signaling in Saragolla, potentially contributing to its inability to achieve complete stomatal closure. We observed a significant increase in *MAPK3/6* in Saragolla wheat, likely due to reduced stromal Ca^2+^ concentration. *ERF109* and *MYB60* play crucial roles in the drought response, with *MYB60*’s downregulation in Svevo leading to a reduced stomatal aperture. Additionally, a higher expression of *TaTPC1* in Svevo suggests its role in accelerating stomatal closure through Ca^2+^ channel activity.

In conclusion, the results indicate that the signaling pathway in the Saragolla alters stomata closure by moderately opening them based on the intracellular calcium concentration associated with the expression of several classes of protein kinases, phosphatases, and plant hormone levels. In the long term, this leads to the preservation of photosynthetic efficiency in drought-stressed wheat, contributing to the interpretation as to why the ancient cultivar Saragolla tends to be more stress-tolerant than the susceptible modern one Svevo. Nevertheless, further molecular and signaling pathway studies might bring us closer to a fuller and more relevant understanding of stomatal action, especially in the ancient variety.

## Figures and Tables

**Figure 1 plants-13-01575-f001:**
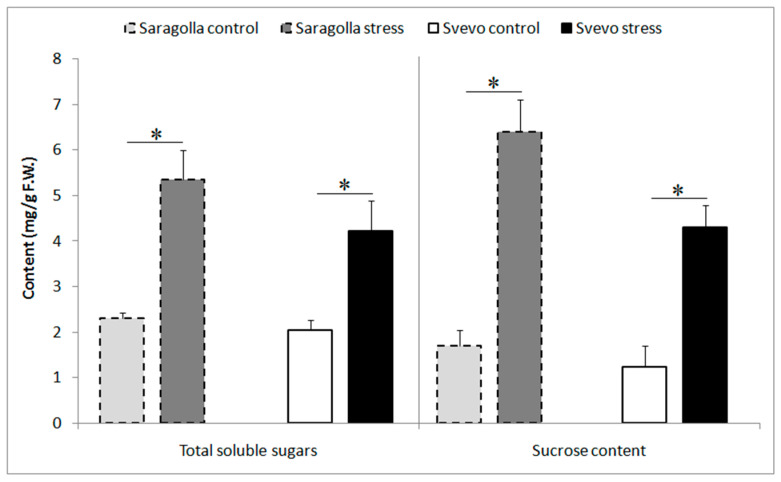
Changes in total soluble sugars and sucrose content (mg/g Fresh Weight) in Saragolla and Svevo leaves after 5 days of osmotic stress treatment with 18% PEG-6000 (*w*/*v*). Data are presented as mean ± SD from at least 3 independent experiments; * *p* < 0.05 (Student’s *t*-test).

**Figure 2 plants-13-01575-f002:**
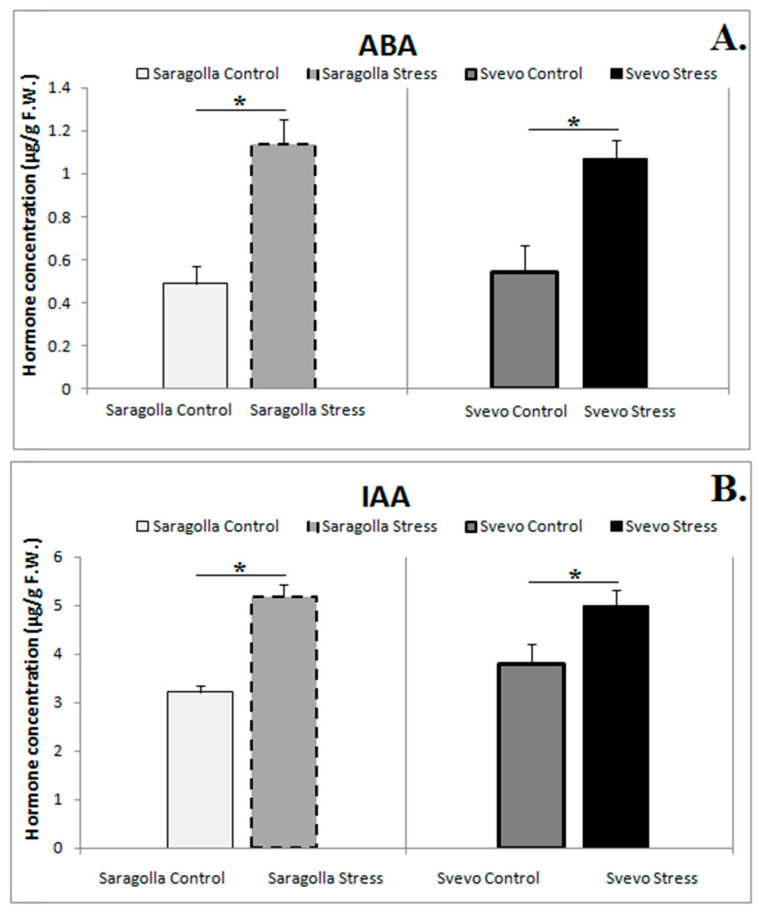
ABA (**A**) and IAA content (**B**) (µg/g Fresh Weight) in Saragolla and Svevo leaves after 5 days of osmotic stress treatment with 18% PEG-6000 (*w*/*v*). Data are presented as mean ± SD from at least 3 independent experiments; * *p* < 0.05 (Student’s *t*-test).

**Figure 3 plants-13-01575-f003:**
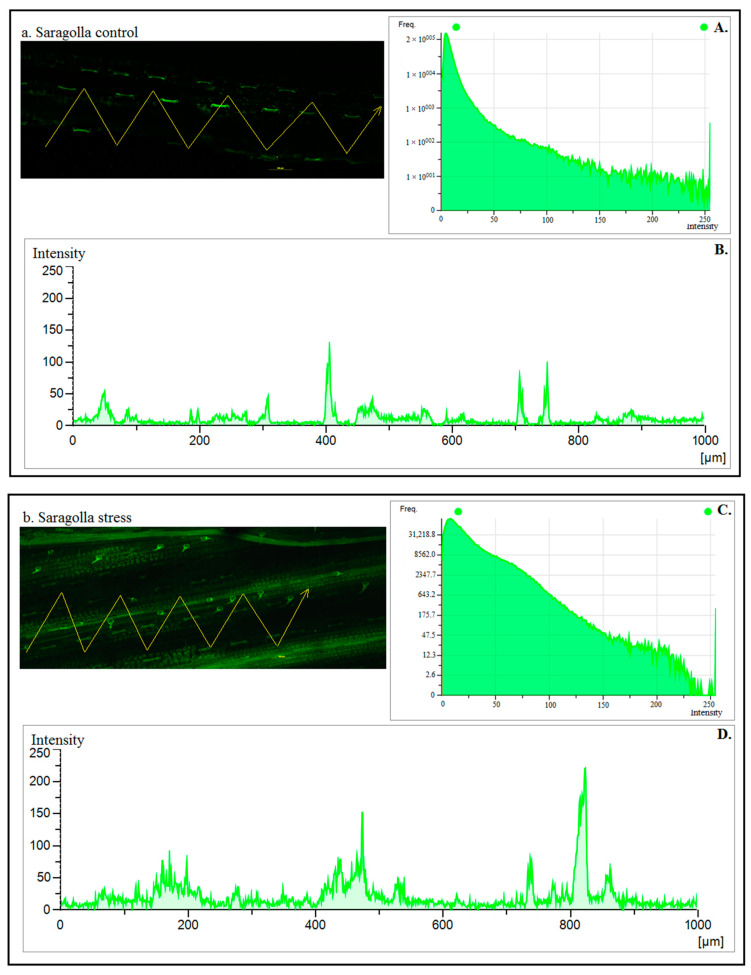
Confocal microscopy images showing the calcium-mobilizing effects in control (**a**,**c**) and PEG-6000 treatment (**b**,**d**) conditions. The pixel intensity–frequency histogram (**A**,**C**,**E**,**G**) and bar charts of the fluorescence intensity (**B**,**D**,**F**,**H**) of seedlings in response to the osmotic stress (**A**,**B**—Saragolla Control; **C**,**D**—Saragolla Stress; **E**,**F**—Svevo Control; **G**,**H**—Svevo Stress). Each pixel on the yellow line has been drawn in (**a**–**d**). Note that under treatment in the Svevo cultivar, there is a significant increase in the intracellular free calcium level [Ca^2+^], with an extremely high intensity of green, compared to Saragolla and the respective control.

**Figure 4 plants-13-01575-f004:**
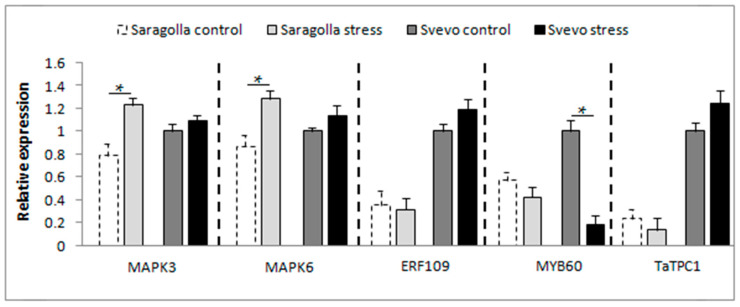
Transcript levels of *MAPK3/6*, *ERF109*, *MYB60*, and *TaTPC1* in leaves of Saragolla and Svevo seedlings under control and osmotic stress determined using RT-qPCR. Data are presented as mean ± SD from at least 3 independent experiments; * *p* < 0.05 (Student’s *t*-test).

**Figure 5 plants-13-01575-f005:**
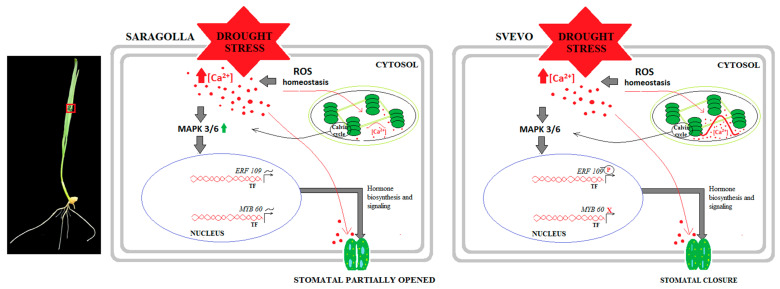
The schematic representation of the hypothetical signaling pathway activated under PEG-6000 treatment in Saragolla and Svevo cultivars. In this pathway, Ca^2+^ concentration in the chloroplast stroma has an important impact on cytosolic *MAPK3/6* phosphorylation by altering stomatal closure. In the Saragolla, a plastid-derived factor is most likely not active, thereby leading to an over-expression of *MAPK3/6*, which alters its kinase activity, leading to the promotion of partial stomatal opening. This eventually results in (almost) unchanged levels of transcription factors *ERF109* and *MYB60*. The image (in the **left**) of the wheat seedling representing the presumed site of the hypothetical signaling pathway (outlined in red) (inspired by Teardo et al. [53]).

**Table 1 plants-13-01575-t001:** Forward and Reverse primer for each study-interested gene.

Genes	Forward Primer	Reverse Primer
*β-ACTIN*	TGGACTCTGGTGATGGTGTC	CCTCCAATCCAAACACTGTA
*ERF 109*	GAGCTACCTCCAGCCATCAC	GCATGTCCAAGGTGTTGTCG
*MAPK 3*	GCGAGGAATCACGGTCTCTT	GATCTGTTGGCGCTTGTTGG
*MAPK 6*	ATCCTGGAATCCTGAGGAGGTT	CCAGGCACAAGCCATCTCAT
*MYB 60*	ACCCGGGATCAAGAGAGGAA	TCTCTCAGCAACACACAGTTC
*TaTCP 1*	TCCCAAAGGGGGATGGTGTT	ATGAGTGGCTTTCCCGCTAC

## Data Availability

The data presented in this article are available on request from the corresponding authors.

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
