# Peer review of "Effect of Polyethylene Glycol-Simulated Drought Stress on Stomatal Opening in “Modern” and “Ancient” Wheat Varieties"

_plants, 2024, doi:10.3390/plants13111575_

Round 1

Reviewer 1 Report

Comments and Suggestions for Authors

The article titled “Effect of Polyethylene Glycol-Simulated Drought Stress on Stomatal Opening in “Modern” and “Ancient” Wheat Varieties” presents the draught stress effect on calcium dynamics as well as the expression of ERF109, MAPK3/6, MYB60, and TaTPC1, involved in the activation of drought-related calcium-sensitive pathways,   in the ancient wheat variety Saragolla and the modern one, Svevo. The study measured and analyzed various biochemical and molecular parameters that can especially condition the stomatal movement and showed a significant increase in the levels of total soluble sugars, ABA, and IAA in both selected cultivars to a greater extent in the Saragolla than in Svevo. The study proposes that the failure of stomatal closure in response to water stress, previously observed in the Saragolla cultivar, is due to a reduced increase in intracellular Ca2+ levels and thus an altered activation of downstream signaling cascades. The study claims that under prolonged stress conditions, these characteristics may help to maintain the photosynthetic efficiency of this wheat variety and, consequently, to maintain high yields even under stress conditions. Therefore, the study asserts the ability of the ancient wheat variety Saragolla to cope better with drought stress than the modern variety, Svevo. The manuscript will be useful for researchers who are interested in draught stress in wheat varieties, ancient and modern ones, and how to tackle the challenges posed by water stress by understanding their molecular mechanisms and their physiological and biochemical effects on wheat crop production. The manuscript is suitable for publication in Plants, however, I have some concerns. The study claims that IAA levels increase significantly in both cultivars under drought stress but should make it clear that the observed increase in IAA might represent a specific response in these cultivars and is not universally observed in all plants. In the study Ca2+ in drought response was measured. However, it notes a "faint fluorescence" in Saragolla indicating a lower calcium concentration under stress. The reduced fluorescence might suggest an issue with the experimental setup or interpretation rather than an inherent property of the cultivar. The study claims that MAPK3/6 levels are activated in Saragolla but not in Svevo under drought stress, which seems inconsistent with the general understanding that stress-responsive MAPKs are typically upregulated under stress conditions across various plant species. The study should provide more context or experimental validation for these results. The study suggests that the inability of Saragolla to fully close its stomata under drought stress is due to altered Ca2+ signaling. This interpretation might be oversimplified. Stomatal behavior is regulated by a complex network of signals, including ABA, Ca2+, reactive oxygen species (ROS), and other secondary messengers. A more comprehensive explanation should be provided to explain how these factors interact in Saragolla. The abstract of the paper should reflect the content of the manuscript. The manuscript needs some English corrections too, including mistakes, spelling, articles, etc.

Comments on the Quality of English Language

Minor editing of the English language is required.

Author Response

Dear Reviewer,

First of all, we wish to thank you for your revision of the paper and for the helpful comments. The points of weakness you spotted have brought us the opportunity to further improve our work. On the basis of your suggestions, we changed and improved some parts of our article. Concerning the other specific comments of the reviewer, below please find the answers.

Comment: The study claims that IAA levels increase significantly in both cultivars under drought stress but should make it clear that the observed increase in IAA might represent a specific response in these cultivars and is not universally observed in all plants.

Answer: Recent bibliographic resources have identified a tangible link between auxin content and plant drought stress response, demonstrating that auxin homeostasis regulates ABA production and drought stress responses (Farhangi-Abriz and Torabian, 2018). Studies suggest that increased IAA concentration promotes water uptake into protoplasts, which is generally associated with stimulated stomatal opening, acting as an antagonist to ABA (Pospíšilová, 2003; Gaion et al., 2018). This pattern was also evident in our results, where IAA concentration increased significantly more in the Saragolla cultivar compared to the other cultivar under PEG-6000 treatment. Given that ABA content increased to similar levels in both treated cultivars, the substantial rise in IAA concentration in Saragolla likely contributed to the partial opening of its stomata (added in line). This response highlights important data regarding stomatal regulation.

Comment: In the study Ca2+ in drought response was measured. However, it notes a "faint fluorescence" in Saragolla indicating a lower calcium concentration under stress. The reduced fluorescence might suggest an issue with the experimental setup or interpretation rather than an inherent property of the cultivar.

Answer: As it is reported in our study case, for both cultivars subjected to osmotic stress, the stress-induced Ca2+ oscillations resulted in fluorescence signal increment. Even though the effect of PEG-6000 on Ca2+ dynamics was more pronounced in the cultivar Svevo than in Saragolla showing a greater change in Ca2+ distribution and a higher increase in fluorescence signal intensity. The aforementioned result piqued our curiosity, prompting us to investigate the underlying cause of the fluorescence change. We wanted to determine whether this change was indeed a result of experimental setup or interpretation as was suspected even by you, or it was a significant response related to the physiological processes of the selected wheat cultivars. Thereby, we focused on the relationship between guard cell closure and intracellular Ca2+ concentration, examining the expression of various protein kinases and phosphatases as signal transducers involved in osmotic stress signaling (Hetherington, 2001; Hetherington and Woodward, 2003; Tuteja and Mahajan, 2007; Brandt et al., 2015; Liu et al., 2022).

Comment: The study claims that MAPK3/6 levels are activated in Saragolla but not in Svevo under drought stress, which seems inconsistent with the general understanding that stress-responsive MAPKs are typically upregulated under stress conditions across various plant species.

Answer: It is reported that MAPKs are typically activated in response to stress, leading to the phosphorylation of downstream targets and the initiation of stress responses. But it is reported as well that after initial activation, there may be feedback mechanisms that reduce the activity or expression of MAPK3 and MAPK6 to prevent overactivation of the stress response, such as the role of some specific phosphatases that can deactivate MAPKs by dephosphorylating them, which effectively downregulates their activity.  At this base,  the significant increase of MAPK3/6 in Saragolla treated reported in our study might be due to the reduced stromal Ca2+ concentration under osmotic stress in the Saragolla cultivar, as MAPKs, despite being activated in the light, are known to be inhibited by high stromal Ca2+ concentrations (Teardo et al., 2019). Another explanation proposed by Teardo et al. (2019) is that a photosynthesis-related metabolite, likely not released in Saragolla under osmotic stress, could lead to the over-phosphorylation of MAPK3/6. However, further investigations are needed to accurately analyze the spatial and temporal distribution of Ca2+ under drought stress and the potential non-release of the photosynthesis-related metabolite.

Comment: The study suggests that the inability of Saragolla to fully close its stomata under drought stress is due to altered Ca2+ signaling. This interpretation might be oversimplified. Stomatal behavior is regulated by a complex network of signals, including ABA, Ca2+, reactive oxygen species (ROS), and other secondary messengers. A more comprehensive explanation should be provided to explain how these factors interact in Saragolla.

Answer: We hypothesized that the partial closure may be linked to alterations in the calcium signal, which are particularly associated with gene expression and hormonal responses. We believe this complex hypothesis could somehow help us clarify our uncertainties regarding the partial opening of the stomata. A better explanation is provided in the section 5. Conclusion, corresponding to:

Stomatal behavior in plants, including wheat cultivars, is regulated by a complex network of signals that interact to manage water loss and gas exchange, particularly under drought stress, such as hormonal cues, Ca2+, reactive oxygen species (ROS), and other secondary messengers. In tolerant wheat cultivars, these signals integrate to modulate stomatal aperture, ensuring water efficiency while minimizing the impact on photosynthesis. Based on that, in summary our results provide further evidence of Saragolla's superior adaptation to drought stress, offering insights into the intricate mechanisms that can inform the development of improved drought-tolerant wheat varieties. This is crucial for ensuring food security in the face of climate change-induced water scarcity.

Our investigation into biochemical parameters revealed a significant increase in soluble sugars and sucrose content under drought stress, with Saragolla showing a more pronounced effect. This indicates that sugars contribute to the improved drought stress tolerance of Saragolla, acting as osmoprotectants and supporting essential cellular functions. Additionally, our data showed a substantial increase in ABA and IAA levels in both cultivars under drought stress, with Saragolla exhibiting a higher percentage increase in IAA concentration compared to Svevo. In the majority of studies this is associated with stimulated stomata opening and is considered an ABA antagonist, likely contributing  to the partial stomata opening observed in Saragolla under drought stress. Furthermore, our study highlights the complex interplay between calcium concentration, MAPK phosphorylation, and transcription factor regulation in stomatal movement under osmotic stress. Differences in calcium concentration between Saragolla and Svevo suggest altered Ca2+ signaling in Saragolla, potentially contributing to its inability to achieve complete stomatal closure. We observed a significant increase in MAPK3/6 in Saragolla wheat, likely due to reduced stromal Ca2+ concentration. ERF109 and MYB60 play crucial roles in drought response, with MYB60 down-regulation in Svevo leading to reduced stomatal aperture. Additionally, higher expression of TaTPC1 in Svevo suggests its role in accelerating stomatal closure through Ca2+ channel activity.

In conclusion, the results indicate that the signaling pathway in the Saragolla alters stomata closure by moderately opening them based on the intracellular calcium concentration associated with the expression of several classes of protein kinases, phosphatases, and plant hormone levels. In the long-term, this leads to the preservation of photosynthetic efficiency in drought-stressed wheat, contributing to the interpretation as to why the ancient cultivar Saragolla tends to be more stress-tolerant than the susceptible modern one Svevo. Nevertheless, further molecular and signaling pathway studies might bring us closer to a fuller and more relevant understanding of stomatal action, especially in the ancient variety.

P.s. Moreover, as you suggested, the manuscript has also undergone extensive English revisions.

We hope the reply provided a satisfactory answer,

Sincerely,

Mariapina Rocco

References

  1. Brandt B, Munemasa S, Wang C, Nguyen D, Yong T, Yang PG, Poretsky E, Belknap TF, Waadt R, 418 Alemán F, Schroeder JI (2015) Calcium specificity signaling mechanisms in abscisic acid signal 419 transduction in Arabidopsis guard cells. Elife 4, e03599.
  2. Gaion, L. A., Braz, L. T., Carvalho, R. F. (2018). Grafting in vegetable crops: a great technique for agriculture. Int. J. Vegetable Sci. 24, 85–102.
  3. Hetherington AM (2001) Guard cell signaling. Cell 107, 711-714. 474
  4. Hetherington AM, Woodward FI (2003) The role of stomatal in sensing and driving environmental 472 change. Nature 424, 901–908.
  5. Farhangi-Abriz S., Torabian S. Biochar Increased Plant Growth-Promoting Hormones and Helped to Alleviates Salt Stress in Common Bean Seedlings. J. Plant Growth Regul. 2018; 37, 591–601.
  6. Liu H, Song S, Zhang H, Li Y, Niu L, Zhang J, Wang W (2022) Signaling Transduction of ABA, ROS, and Ca2+ in Plant Stomatal Closure in Response to Drought. International Journal of Molecular Sciences 23, 14824.
  7. Pospíšilová, J. (2003). Participation of phytohormones in the stomatal regulation of gas exchange during water stress. Biol. Plant 46, 491–506.
  8. Teardo E, Carraretto L, Moscatiello R, Cortese E, Vicario M, Festa M, Szabo I (2019) A chloroplast-localized mitochondrial calcium uniporter transduces osmotic stress in Arabidopsis. Nature Plants 5, 581-588.
  9. Tuteja N, Mahajan S (2007) Calcium signaling network in plants: an overview. Plant signaling & behavior 2, 79-85.

Reviewer 2 Report

Comments and Suggestions for Authors

In this article, Licaj and his (her) colleagues aim to explore the effect of PEG-6000-induced drought stress in the ancient wheat variety Saragolla and the modern one Svevo. The manuscript also provided new evidence regarding the ability of the Saragolla to adapt more favorably to drought stress than the Svevo genotype. The conclusions are supported by the data, and the submitted manuscript is written well and interest to the readers. However, I have several comments that should be addressed before publication.

In scientific aspects:

1.       In the section of Materials and Methods, Please check the sentences “4.5 PCR (qRT-PCR) Analysis”(Page12, L394)  and “……was used for RT-qPCR” (Page12, L404), which are correct? “qRT-PCR” or “RT-qPCR”?  I suggested that the “RT-qPCR” (Page12, L404) should be replaced by “qRT-PCR”.

2.       The authors should give an explain why to analyze the expressions of TaTPC1, ERF109, MYB60, and MAPK3/6 instead of other genes, or why to select them? Moreover, only TaTPC1 be with an abbreviation of Triticum aestivum before the TPC1, why the ERF109, MYB60, and MAPK3/6 are without “Ta”? They are not orthologous genes in the Arabidopsis?

Comments on the Quality of English Language

English is good!

Author Response

Dear Reviewer,

We thank you for your analysis and especially the helpful hints you gave to our work

  1. Comment: In the section of Materials and Methods, Please check the sentences “4.5 PCR (qRT-PCR) Analysis”(Page12, L394)  and “……was used for RT-qPCR” (Page12, L404), which are correct? “qRT-PCR” or “RT-qPCR”?  I suggested that the “RT-qPCR” (Page12, L404) should be replaced by “qRT-PCR”.

Answer: Corrected as you suggested (qRT-PCR).

  1. Comment:The authors should give an explain why to analyze the expressions of TaTPC1, ERF109, MYB60, and MAPK3/6 instead of other genes, or why to select them?

Answer: As it is reported even in the line 58-60: We specifically investigated the expression of the genes ERF 109, MAPK 3/6, MYB60, and TaTPC1, as they play roles in calcium-sensitive response pathways under drought stress and indirectly influence stomatal movement. One possible explanation for the link between calcium concentration and these transcription factors (TFs) involves the stromal Ca2+ -regulated production and/or export of Calvin cycle metabolites to over-phosphorylated MAPK3/6. This process may prevent the activation of ERF106 and MYB60, partially mediating stomatal opening. Additionally, we studied TaTPC1 expression, a gene encoding Ca2+ -permeable channels, which is crucial for facilitating Ca2+ entry into plant cells and affecting stomatal movement (Wang et al., 2005).

  1. Comment: Moreover, onlyTaTPC1 be with an abbreviation of Triticum aestivum before the TPC1, why the ERF109, MYB60, and MAPK3/6 are without “Ta”? They are not orthologous genes in the Arabidopsis?

Answer: Gene-specific primers corresponding to ERF109, MYB60 and MAPK3/6 were designed for Triticum aestivum by us using the NCBI Primer Blast tool. Whereas TaTCP1 primers were already designed and tested by Wang et al., 2005. This was updated even in the manuscript.

We hope the reply provided a satisfactory answer,

Sincerely,

Mariapina Rocco

References

  1. Wang, Y. J.; Yu, J. N.; Chen, T.; Zhang, Z. G.; Hao, Y. J.; Zhang, J. S.; Chen, S. Y. Functional analysis of a putative Ca2+ channel gene TaTPC1 from wheat. Journal of experimental botany, 2005, 56, 422, 3051-3060.